# Prediction of Daily Ambient Temperature and Its Hourly Estimation Using Artificial Neural Networks in Urban Allotment Gardens and an Urban Park in Valladolid, Castilla y León, Spain

Francisco Tomatis [1], Francisco Javier Diez [1], Maria Sol Wilhelm [2] and Luis Manuel Navas-Gracia [1],*

[1] TADRUS Research Group, Department of Agricultural and Forestry Engineering, University of Valladolid, 34004 Palencia, Spain; francisco.tomatis@uva.es (F.T.); x5pino@yahoo.es (F.J.D.)

[2] Centro de Estudios de Variabilidad y Cambio Climático, Facultad de Ingeniería y Ciencias Hídricas, Universidad Nacional del Litoral, Santa Fe 3000, Argentina; msolwilhelm@gmail.com

* Correspondence: luismanuel.navas@uva.es

**Abstract:** Urban green spaces improve quality of life by mitigating urban temperatures. However, there are challenges in obtaining urban data to analyze and understand their influence. With the aim of developing innovative methodologies for this type of research, Artificial Neural Networks (ANNs) were developed to predict daily and hourly temperatures in urban green spaces from sensors placed in situ for 41 days. The study areas were four urban allotment gardens (with dynamic and productive vegetation) and a forested urban park in the city of Valladolid, Spain. ANNs were built and evaluated from various combinations of inputs (X), hidden neurons (Y), and outputs (Z) under the practical rule of "making networks simple, to obtain better results". Seven ANNs architectures were tested: 7-Y-5 (Y = 6, 7, . . ., 14), 6-Y-5 (Y = 6, 7, . . ., 14), 7-Y-1 (Y = 2, 3, . . ., 8), 6-Y-1 (Y = 2, 3, . . ., 8), 4-Y-1 (Y = 1, 2, . . ., 7), 3-Y-1 (Y = 1, 2, . . ., 7), and 2-Y-1 (Y = 2, 3, . . ., 8). The best-performing model was the 6-Y-1 ANN architecture with a Root Mean Square Error (RMSE) of 0.42 °C for the urban garden called Valle de Arán. The results demonstrated that from shorter data points obtained in situ, ANNs predictions achieve acceptable results and reflect the usefulness of the methodology. These predictions were more accurate in urban gardens than in urban parks, where the type of existing vegetation can be a decisive factor. This study can contribute to the development of a sustainable and smart city, and has the potential to be replicated in cities where the influence of urban green spaces on urban temperatures is studied with traditional methodologies.

**Keywords:** urban temperature; urban climate; urban gardens; urban parks; urban green spaces; urban climate mitigation; artificial neural networks; predictions

## 1. Introduction

Urban climate research and the influence of urban green spaces on temperature mitigation is a topic of interest [1–3], especially as cities face the challenge of increasing populations and threats of climate change. Currently, more than half of the world's population lives in cities, and this is expected to rise to 60% by 2030 [4]. Consequently, pursuing more sustainable, habitable, smart, and resilient cities are the most significant challenges for urban policy and planning in the 21st century [4,5].

The term "urban" encompasses a wide range from rural towns to megacities [6]. Nevertheless, medium-sized cities and megacities, in particular, exhibit significant spatial heterogeneity [7], along with lower albedo, higher thermal conductivity, and the highest heat capacities of building materials [8,9]. As a result, the urban climate represents a distinctive microclimate that emerges from the combined influence of buildings and human activities, significantly impacting the thermal energy dynamics of the city [10]. The Urban Heat Island (UHI) effect can be defined as the development of higher temperatures in the city's center compared to adjacent rural areas [10,11] and is currently a cause of concern [12].

Many studies confirmed that as the proportion of impervious surfaces increases within an urban area, the UHI effect becomes more pronounced [9,13–15].

In this context, particularly in climate change scenarios, it has been identified that urban areas are not sufficiently adapted to global warming [16]. The UHI effect exacerbates the already warm conditions experienced in cities [17–20], leading to increased exposure of people to higher heat [21] and negatively affecting their quality of life [22]. Although urban green spaces are scarce and threatened [7,23,24], their vegetation cover can mitigate temperatures and heat waves [21,25–27], potentially making them valuable solutions for urban climate adaptation [28]. It was shown that urban gardens, particular green spaces due to their dynamic and productive vegetation, can improve the mitigation of urban temperatures [21,28–30].

To empirically understand what happens in these urban spaces, researchers in urban climatology have highlighted the need for reliable meteorological data in the local urban context [31]. Generally, the investigations related to the effect of green spaces, such as urban gardens and urban parks, on urban temperatures are based on methods and analyses that involve in situ sensors for long periods (for example, during all the summer) without taking advantage of the innovative predictive capabilities of Artificial Neural Networks (ANNs) [21,29,30,32]. Based on their results and practicality, ANNs offer a valuable tool to understand and address the complex dynamics of urban climates [31,33,34]. However, there is a lack of scientific evidence that develops ANNs for urban temperature focusing on the influence of urban green spaces, such as urban gardens and urban parks, from in situ data.

For this purpose, the prediction of daily ambient temperature and its hourly estimation in four urban allotment gardens and in an urban forested park were developed in this study using ANNs. The results obtained demonstrate the usefulness and validity of the ANNs, since from short data points obtained in situ (41 days), interesting urban estimates and predictions can be made. The research is innovative because it incorporates the use of available technologies for this type of studies that are often developed using traditional methodologies. Although better results can still be sought, the research developed leaves solid evidence for its optimization. The study stimulates the development of ANNs for urban temperatures, supports the influence of urban green spaces on the mitigation of temperatures, contributes towards the management of a sustainable and smart city, and leaves a basis for replication in other cities and studies.

*Artificial Neural Networks (ANNs) for Urban Temperatures*

To determine urban temperatures, data from meteorological weather stations located primarily in peri-urban and rural areas (such as airports) are often considered. As the UHI effect intensifies, temperature data from peri-urban and rural areas become less representative of the current urban climate. For this reason, one alternative for determining urban temperatures are to instal in situ sensors throughout the city. Nevertheless, not all municipalities are willing to invest in this approach due to its economic cost and the risks of theft or damage, which limit its suitability for long-term studies [31,35].

Within this framework, some of the most important advances are concentrated in the modeling field [36] where various artificial intelligence techniques, including ANNs, genetic algorithms, fuzzy logic, and hybrid technique approaches, are developed to estimate ambient temperature [37,38]. The ANNs are an innovative alternative that are distinguished by their ability to exploit unknown and hidden information in climate data, which may not be directly extractable [38,39]. Some studies demonstrate the use of ANNs and are oriented to the estimating and predicting temperatures for countries [40], for regions [37,41–43], and others for cities [28,36,44–53]. The cities identified where these studies have been carried out include Montreal (CA), Vancouver (CA), Stuttgart (DE), Austin (US), Hong Kong, Karachi (PK), Athens (GR), Madrid (ES), Rotterdam (UN), Seoul (KR) and Abu Dhabi (AE), among others [28,34,46,47,49–53].

In the case of cities, the identified bibliographies present characteristics that allow them to be compared to this study due to their approaches, objectives, and methodologies for the development of ANN focus on urban temperatures. The interest of these bibliographies in developing ANNs for urban temperatures is related to UHI effect [28,36,44,45,48,49,52], thermal comfort [46,51], heat stress [47], land surface temperature [43,53], and building energy [45]. Studies have also found that developing ANNs can relate urban temperatures to urban green spaces, such as urban parks [51] and green roofs [52]. In the case of the urban park, the study focused on predicting the thermal comfort of people [51].

It is important to consider that the periods of recording of temperature data, or other climate variables, used for ANNs inputs in identified studies can range from data from years [28,43,44,53], months [45,46,48], days [47], and even data for less than 24 h [49]. The source of this data is usually combined or separate from traditional meteorological weather stations, urban sensors placed in situ or on urban weather stations, and sensing satellites [36,43–47,52–54].

Franco et al. [55] found that there is a lack of such studies that use ANNs models and that focus on generating data in places where such data is not available so that they can be used as inputs for other models. In this study, it was identified that there is a lack of scientific evidence that develops ANNs for urban temperatures, mainly focusing on urban green spaces such as urban gardens and urban parks from data obtained in situ for short periods of time.

In addition, there are also other alternatives used to make urban temperature predictions; for example, IoT, which is able to predict the citywide surface temperatures with only ambient sensors on board [56], and statistical multivariate methods, which predict the external temperature value of urban road channels [57], or even remote sensing observation or computer models, which are used to retrieve the spatial distribution of UHI [58]. Each approach offers unique advantages and considerations for urban temperature prediction.

## 2. Materials and Methods

This section outlines the following steps. Section 2.1, the study area; Section 2.2, the ambient temperature data considered for the ANNs; Section 2.3, the ANNs models to estimate urban temperatures; and Section 2.4, the statistics used to analyze the ANNs results.

### 2.1. The Study Area

The city selected is Valladolid, Castilla y León, Spain (Figure 1). Valladolid was chosen because research on urban gardens and climate change is being implemented there based on the interest of the University of Valladolid and the Valladolid City Council. This city offers an ideal environment to investigate the impacts of urban allotment gardens and urban parks on urban temperatures with ANNs, because it is a medium-sized city with aspirations in incorporating nature-based solutions to improve their climate adaptation [59] and to transform itself into a smart city [60].

Valladolid is located in the northwestern quadrant of the Iberian Peninsula, in Spain [61]. On 1 January 2022, it had a population of 295,639 according to Spain's National Statistics Institute [61].

Based on the Climate Change Adaptation Strategy of Valladolid [61], the city predominantly lies in a flat area, and its climate is categorized as Mediterranean-Continental, characterized by an annual rainfall of approximately 400 mm/year and an average temperature that barely exceeds 12 °C. However, climate models predict significant changes due to increased emissions, resulting in considerable alterations in temperature trends. The worst-case scenarios (RCP8.5 and RCP4.5) for Valladolid in the future indicate the following temperature changes [61]:

- A decrease of minimum temperature; extreme minimum temperature; the number of frost days; and the heating degree days;

- An increase of maximum temperature; extreme maximum temperature; thermal amplitude; the number of warm days; the number of heat waves days; and the cooling degree days.

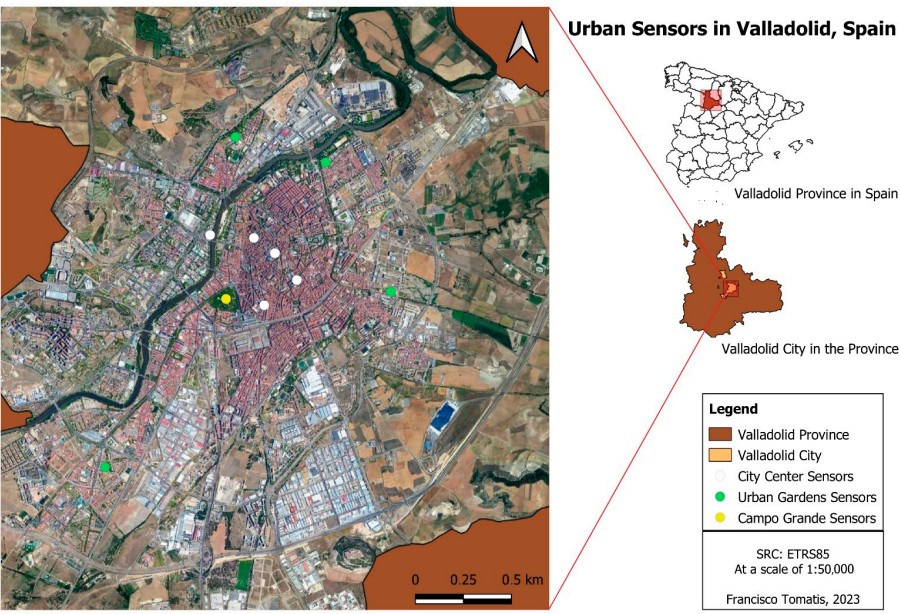

**Figure 1.** City of Valladolid, Spain.

The urban green spaces from Valladolid considered in this study are four urban allotments gardens called Jardín Botánico, Valle de Arán, Los Santos-Pilarica and Parque Alameda, and an urban forested park called Campo Grande.

According to the Call for Ecological Urban Gardens 2022–2023 of the City of Valladolid, N° 10677/2021 [62], the urban allotment gardens spaces are for unemployed people and for organic production. These four areas have different characteristics:

- Jardin Botánico: 33 plots of 50 m$^2$ in a total surface area of 2600  m$^2$;
- Valle de Arán: 50 plots of 50 m$^2$ and 1 plot for community work of 800 m$^2$, in a total surface area of 4620 m$^2$;
- Parque Alameda: 48 plots of 50 m$^2$ and 1 plot for community work of 300 m$^2$, in a total surface area of 3300 m$^2$;
- Los Santos Pilarica: 50 plots of 50 m$^2$ and 1 plot for community work of 800 m$^2$, in a total surface area of 5200 m$^2$.

On the other hand, Campo Grande has a total surface area of 101,376 m$^2$. Campo Grande is the urban park in Valladolid with the highest density of trees, with almost 90 different species [63,64].

Through field work carried out by the University of Valladolid on 24 June 2022 and 29 June 2022 on the urban allotment gardens, the crops present were identified, and their height was measured. Three plots for each garden were selected. As general results, onions, tomatoes, lettuces, peppers, beans, cucumbers, carrots, cabbages, eggplants, garlics, chards, and others complementary crops are cultivated. Regarding the height of the crop vegetation in the study plots, an average of 48 cm was recorded in Jardin Botánico, 43 cm in Valle de Arán, 33 cm in Parque Alameda, and 39 cm in Los Santos Pilarica. This field work is available at Supplementary Materials and reflects the differences in vegetation between the crops of urban allotment gardens and the trees and shrubs of Campo Grande.

### 2.2. The Ambient Temperature Data Considered for the ANNs

The urban temperature data for Valladolid were obtained from the period of 21 June 2022 to 18 July 2022 for ANNs models training and from 19 July 2022 to 31 July 2022 for

ANNs models validation. The urban temperature data in Valladolid were collected using twelve sensors from the University of Valladolid and using five nanosensors operated by the Valladolid City Council.

The loggers from the University of Valladolid were Onset HOBO UA-002-64 (5.8 cm × 3.3 cm × 2.3 cm in size, https://www.onsetcomp.com/products/data-loggers/ua-002-64, accessed on 12 December 2023) with an operating range of −20 to 70 °C in the air, and an accuracy of ±0.53 °C from 0 to 0–50 °C. The selection criteria for sensor type, quantity, installation height (1.5 m), and their locations, followed methodologies used, validated and recommended by researchers who are dedicated to investigating the impact of urban gardens on urban temperatures using traditional methodologies [29,30]. In this context, three sensors were deployed in each urban garden (Figure 2) due to the diverse vegetation within the same area and two sensors were placed in Campo Grande (Figure 3), given its more spatially and temporally uniform vegetation (this characteristic potentially results in lower variability of temperature fluctuations at multiple points within the same area).

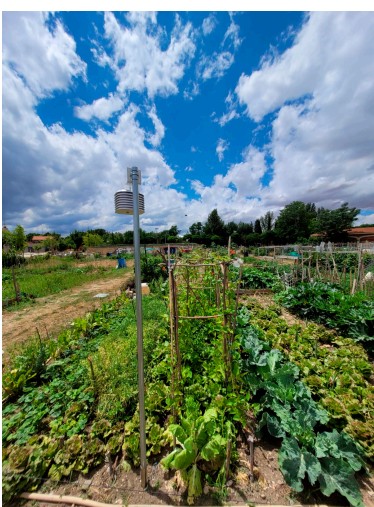

**Figure 2.** Sensor located in the urban allotment garden called Valle de Arán.

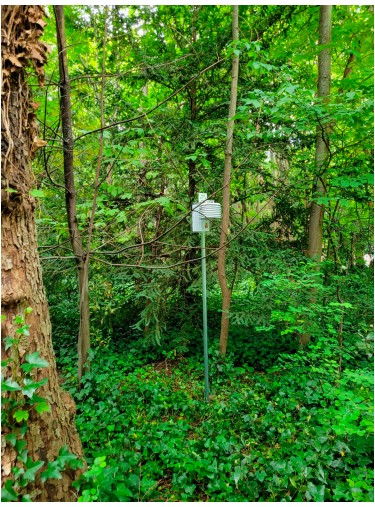

**Figure 3.** Sensor located in the forested urban park called Campo Grande.

Furthermore, data was collected from five nanosensors operated by the Valladolid City Council, situated in downtown places without vegetation namely Puente Poniente, Plaza San Miguel, Catedral, Don Sancho and Dos de Mayo.

The initial data obtained from the sensors from the University were hourly temperature readings, while the data from the Valladolid City Council were recorded every 15 min. These readings were converted into average hourly temperatures for further analysis with the ANNs. The average hourly values of each of the locations were taken into consideration because they provide more representative and stable input data for models training, encouraging better generalization and performance.

For the process of standardizing the different sensors, the University of Valladolid's equipment was positioned in the same places as the sensors belonging to the Valladolid City Council, a process referred to as calibration. This calibration procedure spanned a total of 84 h, commencing at 00 a.m. on 21 June 2022 and concluding at 11 a.m. on 24 June 2022. Subsequently, the values obtained from the sensors, synchronized with the official time of Spain, were adjusted to solar time (−2.5 h).

After the time correction, an additional step was taken to homogenize the data obtained from both sets of sensors. This homogenization was achieved using a scaling factor, defined as the ratio between the average hourly value observed by the sensors and the average hourly value observed by the Council sensors at each of the calibration sites. From this, individual scaling factors were determined for each of the five calibration sites, and an overall scaling factor calculated as the average of these five individual scaling factors (F = 1.04). This overall scaling factor was then applied to each dataset obtained from the urban allotment gardens and Campo Grande. Through this approach, the data was effectively corrected based on time, resulting in a significant reduction in the disparities between the two types of sensors.

### 2.3. The ANNs Models to Estimate Urban Temperatures

The developed ANNs architectures (X-Y-Z) changed the number of inputs (X), neurons in the hidden layers (Y), and outputs (Z). From this, various models were built and evaluated using different combinations of inputs, hidden neurons, and outputs, following the practical guideline of "making networks simple, to obtain better results". The seven ANNs architectures developed were: 7-Y-5, 7-Y-1, 6-Y-5, 6-Y-1, 4-Y-1, 3-Y-1, and 2-Y-1.

The model configurations were guided by Occam's razor principle, prioritizing simplicity for optimal performance in ANNs. "Occam's razor" is a general guideline in science, emphasizing that the most straightforward is often the most probable, though not necessarily the true one. We began our tests with configuration 7-Y-5, X = (5 Valladolid City Council Data + Day of the Year + Hour of the Day), due to its prior success in solar prediction [38]. However, in our shorter and seasonally restricted dataset, this complexity did not yield better results compared to a simpler model, 6-Y-5. Similar trends were observed when comparing 7-Y-1 with 6-Y-1, where the former consistently outperformed the latter by reducing the number of outputs. Likewise, when comparing 3-Y-1 and 4-Y-1 with 2-Y-1, simplifying the model without Day of the Year consistently yielded better results, except when reducing inputs did not lead to satisfactory outcomes.

The ANNs were implemented using the 'feedforwardnet' function in MATLAB. The dimensions of the input and output data vectors determined the size of the respective layers, resulting in a multilayer feed-forward (MLP) perceptron with a single hidden layer. The activation function chosen between neurons in the hidden layer was the hyperbolic sigmoid tangent (tansig), while the transfer function for neurons in the output layer was linear (purelin). The Levenberg–Marquardt back-propagation algorithm (BP-LM) was applied to achieve fast optimization (trainlm) [37]. This commonly used ANNs type and the MATLAB platform are prevalent in this field. While there's potential for improving interpolation performance with different ANNs types, our current emphasis is on finding the most suitable (X-Y-Z) architecture with minimal data requirements.

The ANNs were trained using the ´train´ function, using matrices of input and output data vectors covering 28 days (from 21 June 2022 to 18 July 2022). Following the training phase, the ANNs were tested using the 'sim' function. This testing involved evaluating the previously trained ANNs with various numbers of neurons in the hidden layer (1, 2, . . ., 14

depending on the case). The estimations were made separately for each urban garden and the forested urban park covering 13 days (from 19 July 2022 to 31 July 2022). This dataset served as a reference for validation.

This architecture of the models evaluated is illustrated in Figure 4, featuring six inputs: hourly ambient temperature from five downtown places without vegetation (Puente Poniente, Plaza San Miguel, Catedral, Don Sancho and Dos de Mayo) and one from the Time of the Day (0, 1, …, 23). The output layer was designed with a single neuron, enabling the estimation of ambient temperatures for each urban garden and for Campo Grande, considering the specific Time of the Day.

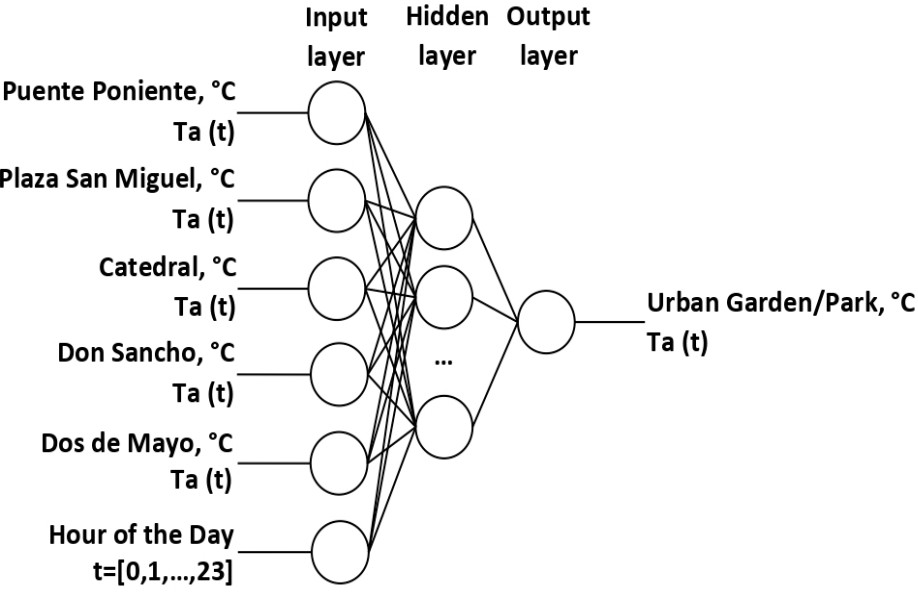

**Figure 4.** 6-Y-1 ANN architecture provided the best results for predictions.

*2.4. The Statistics Used to Analyze the ANNs Results*

The accuracy of the ANNs models in the validation stage was achieved by analyzing statistics values, which include: the Root Mean Square Error (RMSE, temperature °C) using Equation (1); the Coefficient of Determination ($R^2$), as an indicator of the level of model fit, using Equation (2), the Durbin–Watson Coefficient (DW), used to detect first-order self-correction between the data, using Equation (3); the Mean Percentage Error (MPE), which allows the interpretation of the bias in the prediction error, employing Equation (4); and the Forecast Accuracy (FA), which is used in short-term forecasting models, using Equation (5).

$$\text{RMSE}(°C) = \sqrt{\frac{\Sigma_{i=1}^{n}(Y_i - \hat{Y}_i)^2}{n}} \tag{1}$$

$$R^2 = 1 - \frac{\Sigma_{i=1}^{n}(Y_i - \hat{Y}_i)^2}{\Sigma_{i=1}^{n}(Y_i - \overline{Y})^2} \tag{2}$$

$$\text{DW} = \frac{\Sigma_{i=1}^{n}\left((Y_i - \hat{Y}_i) - (Y_{i+1} - \hat{Y}_{i+1})\right)^2}{\Sigma_{i=1}^{n}(Y_{i+1} - \hat{Y}_{i+1}))^2} \tag{3}$$

$$\text{MPE} = \frac{\Sigma_{i=1}^{n}\left(\frac{(Y_i - \hat{Y}_i)}{Y_i}\right)}{n} \tag{4}$$

$$\text{FA} = \frac{\Sigma_{i=1}^{n}\left(1 - \left|\frac{(Y_i - \hat{Y}_i)}{Y_i}\right|\right)}{n} \tag{5}$$

## 3. Results

This section presents the results obtained using the different ANNs models created for the prediction of daily ambient temperature and its hourly estimation in four urban allotment gardens and in Campo Grande.

According to the results given by the statistical values, the most effective ANN for estimating urban temperatures was identified as the 6-Y-1 ANN architecture (Table 1). The best values obtained in the 6-Y-1 ANN architecture (ANN 6-7-1 for Valle de Arán) were RMSE = 0.42 °C and $R^2$ = 0.996 (Figure 5). On the other hand, the worst forecast results for the same architecture, 6-2-1 ANN and 6-3-1 ANN for Campo Grande, were RMSE = 1.20 °C and $R^2$ = 0.956 (Figure 6). Considering all the ANNs models developed and evaluated, a range of values was obtained, varying from RMSE = 0.42 °C and $R^2$ = 0.996 (best values achieved) to RMSE = 4.58 °C and $R^2$ = 0.281 (worst values obtained). These are demonstrated in the 6-Y-1 ANN architectures (output for Valle de Arán: 6-7-1 ANN) and in the 7-Y-5 ANN architectures (output for Campo Grande: 7-11-5 ANN), respectively. It was detected that the outputs performed for Campo Grande have the worst results than the outputs obtained for the urban allotment gardens.

**Table 1.** Prediction models for average hourly temperatures (°C) in urban gardens and forested urban park in Valladolid (Spain), according to the different ANNs architectures (X-Y-Z) = (6-Y-1), changing the number of neurons in the hidden layers Y = (2, 3, . . .,8). Where the inputs are X = [Ta(t) Puente Poniente, Ta(t) Plaza San Miguel, Ta(t) Catedral, Ta(t) Don Sancho, Ta(t) Dos de Mayo, Hour of day (t = 0, 1, . . ., 23)], and the output is Z = [Ta(t) Jardín Botánico], Z = [Ta(t) Valle de Arán], Z = [Ta(t) Los Santos-Pilarica], Z = [Ta(t) Parque Alameda], and Z = [Ta(t) Campo Grande] in each case. Adjustment of statistical values.

| Statistics | ANN 6-2-1 | ANN 6-3-1 | ANN 6-4-1 | ANN 6-5-1 | ANN 6-6-1 | ANN 6-7-1 | ANN 6-8-1 |
|---|---|---|---|---|---|---|---|
| **Outputs for Jardín Botánico** | | | | | | | |
| RMSE (°C) | 0.72 | 0.61 | 0.60 | 0.65 | 0.64 | <u>0.60</u> | 0.61 |
| $R^2$ | 0.988 | 0.991 | <u>0.992</u> | 0.990 | 0.991 | <u>0.992</u> | <u>0.992</u> |
| **Outputs for Valle de Arán** | | | | | | | |
| RMSE (°C) | 0.53 | 0.46 | 0.46 | 0.50 | 0.44 | <u>0.42</u> | 0.46 |
| $R^2$ | 0.993 | 0.995 | 0.995 | 0.994 | 0.995 | <u>0.996</u> | 0.995 |
| **Outputs for Los Santos-Pilarica** | | | | | | | |
| RMSE (°C) | 0.77 | 0.73 | 0.72 | <u>0.60</u> | 0.61 | 0.62 | 0.60 |
| $R^2$ | 0.986 | 0.988 | 0.988 | <u>0.992</u> | 0.991 | 0.991 | <u>0.992</u> |
| **Outputs for Parque Alameda** | | | | | | | |
| RMSE (°C) | 0.69 | 0.61 | 0.63 | 0.64 | 0.63 | 0.63 | <u>0.61</u> |
| $R^2$ | 0.989 | <u>0.991</u> | <u>0.991</u> | 0.990 | <u>0.991</u> | 0.990 | <u>0.991</u> |
| **Outputs for Campo Grande** | | | | | | | |
| RMSE (°C) | 1.20 | 1.20 | 1.15 | 1.10 | 1.14 | 1.07 | <u>0.96</u> |
| $R^2$ | 0.956 | 0.956 | 0.960 | 0.963 | 0.960 | 0.965 | <u>0.972</u> |

Architecture of the model based on the number of neurons in each layer (Input–Hidden–Output). The best architectures (results) obtained for each ANN model with their respective statistical variables of interest are shown/underlining identifies the best result found.

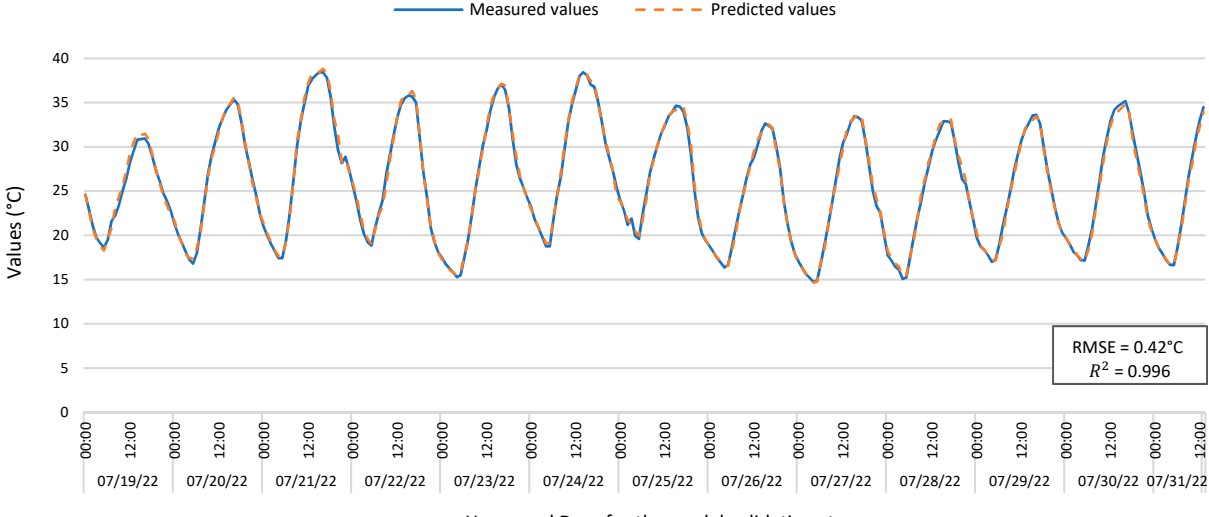

**Figure 5.** Measured data and predictions in Valle de Arán for the validation stage, obtained with the 6-7-1 ANN with the best forecast results of the 6-Y-1 ANN architecture.

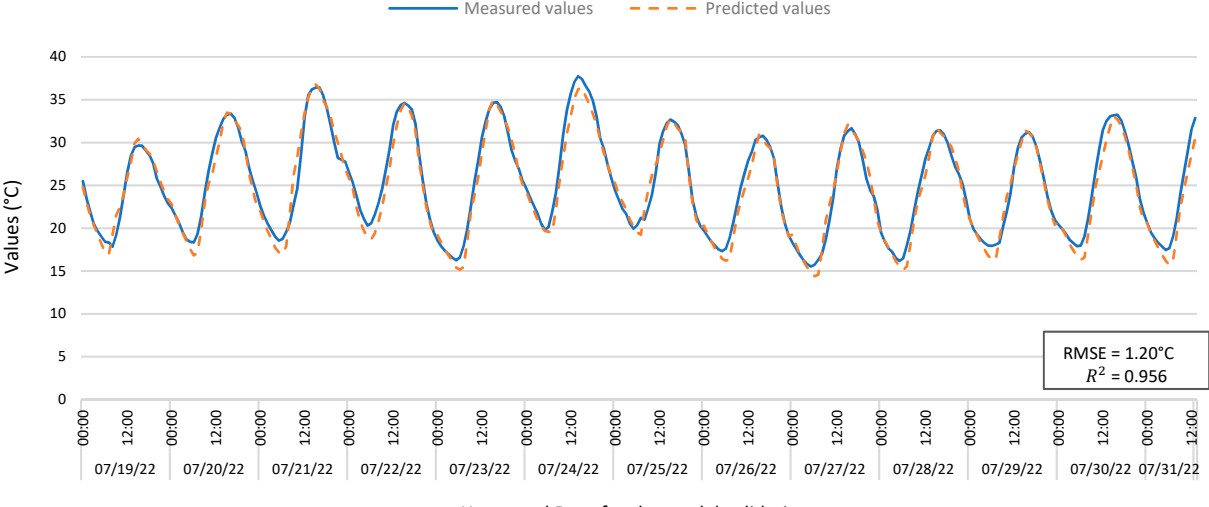

**Figure 6.** Measured data and predictions in Campo Grande for the validation stage, obtained with the 6-2-1 ANN with the worst forecast of the 6-Y-1 ANN architecture.

The 6-Y-1, 7-Y-1, 6-Y-5 and 7-Y-5 ANNs architectures are deemed the most significant due to their outcomes and their applicability. The 4-Y-1, 3-Y-1 and 2-Y-1 ANNs architectures were performed as special tests in search of better results.

The outcomes of all the developed ANNs, along with their corresponding identifications, architectures, and main statistical values are presented in the Supplementary Materials.

## 4. Discussion

In all the ANNs developed, as in the 6-Y-1 ANN architecture, the worst results were found in the predictions for Campo Grande. This is perhaps mainly attributed to the type or amount of vegetation (permanently forested versus dynamic and productive crops). Although a complete sampling of the existing vegetation in the places of study was not carried out, field sampling in some plots of the urban gardens on the date of study easily demonstrates the differences in vegetation between the urban crops in the gardens and a park with trees. The current manuscript does not specifically address how the dynamics of vegetation and urban green areas influence local temperature, but it is part of and

contributes to a more comprehensive study where the relationships between urban gardens and climate change are being studied. Therefore, the methodology developed by this study allows the incorporation of innovative techniques to be incorporated into traditional and specific research on the subject.

Results reveal that the 6-Y-1 ANN architecture outperformed the other architectures developed when the input Day of the Year was eliminated. This improvement was observed in all cases where the ANNs input did not include the Day of the Year variable, such as in 7-Y-1 ANN. Initially, the Day of the Year variable was incorporated in these tests to situate the ANNs in the time of the year. Surprisingly, the inclusion of the Day of the Year variable worsened the results for this specific test, which was conducted over a short period. These findings contradict the results reported in Diez et al. [37], where daily predictions were made for a longer period (one year). In that study, the addition of the Day of the Year variable improved the ANN results. This suggests that the addition of the Day of the Year variable is beneficial for predictions in longer data series, where seasonal patterns have a significant influence. However, in a series of only a few days, where the seasonal variable does not exert a significant influence, the inclusion of the Day of the Year variable did not improve the results. This discrepancy underscores the importance of considering the appropriate input variables depending on the specific context and the length of the data series under examination. The findings highlight the complexity of ANNs models and the need for careful selection and customization of input variables to optimize predictive performance for different scenarios.

The study shows that the parameters used in the ANNs optimize their performance. Nevertheless, the application of certain algorithms can enhance their performance further, opening avenues for future research. For example, additional climate information (solar radiation, humidity, evaporation, among others) from other data sources, such as meteorological weather stations or satellite data, could be added to future studies.

In comparison with other research developed by ANNs for urban temperatures [43–54], this study stands out for considering the influence of urban green spaces on urban temperatures, as there is a lack of scientific evidence linking these topics. This study also stands out because it uses data obtained in situ in an urban area avoiding the use of data from traditional weather stations that are in peri-urban or rural areas. The use of 41 days of data for the development of the ANNs allows for the assumption that if more days are considered, the accuracy could be increased and the ANNs would be more effective. Nevertheless, the results obtained are considered valid since other studies of a similar nature developed this methodology including data from fewer days.

The models carried out in Valladolid could be applicable to other cities, especially when data of a similar nature are available. This possible application in other cities opens the door to the incorporation of innovative tools such as ANNs to analyze the impact of urban gardens on the mitigation of urban temperatures without the need to deploy in situ sensors for long periods of time (such as a summer season). These recommendations could be applied and replicated in cities in the US, Australia, and Germany, for example, where the influence of urban gardens on urban temperatures has been studied [21,29,30]. The ANNs models presented could be applied to data obtained during longer periods (data from an entire station, monthly data or annual data). This research can also be developed in other cities where ANNs are developed for urban temperatures and that do not consider the influence of green spaces. However, it is important to emphasize that the application of the methodology using ANNs would be exported, rather than using the same ANNs for each city, as the ANNs are tailored to the specific data from where they were trained. The method developed with ANNs creates a unique model for each city where it is applied.

## 5. Conclusions

Urban green spaces, such as urban gardens and urban parks, are effective measures to mitigate temperatures and the UHI phenomenon, particularly in the face of demographic and climatic shifts in the 21st century. To study this phenomenon, this article used an

innovative methodology, such as ANNs as an alternative to the existing issues for its measurement, analysis and research.

This research successfully employed ANNs for estimating and predicting temperatures within four urban allotment gardens and one urban forested park in Valladolid. The models successfully generated acceptable results by utilizing in-situ data collected over a period of 41 days, which stands out as an outstanding aspect. Within the seven ANNs architectures tested, the 6-Y-1 ANN architecture emerged as the most effective, notably achieving the lowest Root Mean Square Error (RMSE) value in the Valle de Arán (RMSE = 0.42 °C). This type of prediction was more accurate in urban gardens than in urban parks, where vegetation variation can be a decisive factor.

This study demonstrates that ANNs can provide crucial temperature data without the need to deploy sensors in situ for long periods of time, which is time-consuming, costly, and often carries inherent risks. This innovative methodology allows to change the traditional ways of developing this type of studies and allows it to be replicated in other cities that have data of similar nature or are studying these topics. The promising results, which may even be improved, may help understand the impact of urban gardens on urban climate mitigation, urban agriculture planning, and encouraging adoption of ANNs in urban environments. Furthermore, they contribute towards a more sustainable city by supporting the potential of urban green spaces and a smart city through the application of technologies such as ANNs.

**Supplementary Materials:** The following supporting information can be downloaded at: https://www.mdpi.com/article/10.3390/agronomy14010060/s1 and https://doi.org/10.6084/m9.figshare.24155559.v1, accessed on 18 December 2023.

**Author Contributions:** Conceptualization: F.T., F.J.D. and L.M.N.-G.; methodology: F.T. and F.J.D.; software: F.T. and F.J.D.; validation: F.J.D., M.S.W. and L.M.N.-G.; formal analysis: F.J.D., M.S.W. and L.M.N.-G.; investigation: F.T. and F.J.D.; resources: L.M.N.-G.; data curation: F.J.D. and M.S.W.; writing—original draft preparation: F.T.; writing—review and editing: F.T., F.J.D., M.S.W. and L.M.N.-G.; visualization: F.T. and F.J.D.; supervision: M.S.W. and L.M.N.-G.; project administration, L.M.N.-G.; funding acquisition, L.M.N.-G. All authors have read and agreed to the published version of the manuscript.

**Funding:** This research was funded by the European Union supporting this work through the FUSILLI project (H2020-FNR-2020-1/CE-FNR-07-2020) and the CIRAWA project (HORIZON-CL6-2022-FARM2FORK-01). Francisco Tomatis has been financed under the call for University of Valladolid 2020 predoctoral contracts, co-financed by Banco Santander.

**Data Availability Statement:** Data from field work (crops vegetation in the urban gardens) and the data used for the development of the ANNs, which support the reported results, are available at: https://doi.org/10.6084/m9.figshare.24849267.v1, accessed on 18 December 2023.

**Conflicts of Interest:** The authors declare no conflict of interest.

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
