# Peer review of "Prediction of Daily Ambient Temperature and Its Hourly Estimation Using Artificial Neural Networks in Urban Allotment Gardens and an Urban Park in Valladolid, Castilla y León, Spain"

_agronomy, doi:10.3390/agronomy14010060_

Round 1
Reviewer 1 Report (Previous Reviewer 1)
Comments and Suggestions for Authors
The paper entitled 'Prediction of Daily Ambient Temperature and Its Hourly Estimation Using Artificial Neural Networks in Urban Allotment Gardens and in an Urban Park in Valladolid, Castilla y León, Spain,' submitted for potential publication in Agronomy, delves into a promising field. The use of ANN models for forecasting urban temperatures is indeed a topic of great interest. However, for the manuscript to reach its full potential, I recommend addressing the following points for clarity and completeness.
Introduction:
Reference to 'Significant Attention' (Line 36-39): The claim about the topic receiving significant attention seems unsubstantiated. Could references be provided to support this?
Structure regarding ANNs (Section 1.1): It appears more logical to discuss the broader context of urban temperature variations in Section 1.1 before introducing the specifics of ANNs.
Temperature Models Beyond ANNs (Section 1.2): The discussion currently focuses narrowly on ANNs. A broader perspective on other methodologies for predicting urban temperatures would enrich the section.
Clear Articulation of Contributions: The manuscript should explicitly state its novel contributions, objectives, and hypotheses, both in the abstract and in the body, to better frame the study's importance and direction.
Materials and Methods:
Justification for Valladolid: The rationale for selecting Valladolid as the study location needs to be clearer. Is there specific meteorological data that underscores its suitability?
Sensor Location Criteria (Lines 168-169): The decision-making process behind the sensor placements is unclear. An explanation of the criteria used would be helpful.
Temperature Data Analysis Choice (Line 183): The preference for using average temperatures over maximum temperatures within each hour should be justified.
Detailed Description of Urban Allotment Gardens: Given the significant role of vegetation in influencing temperature through factors like albedo and evapotranspiration, a comprehensive description of the urban allotment gardens studied is essential.
Results, Discussion, and Conclusion:
Reorganization of Tables: To enhance readability, consider relocating Tables 1, 2, and 3 to an appendix. The main text should then focus on distilling and discussing the key results.
Vegetation Dynamics and Temperature Influence: The current manuscript does not adequately address how vegetation dynamics influence temperature. A detailed analysis of this aspect, particularly with the limited dataset used, is crucial.
In its current state, the manuscript, while intriguing, does not fully demonstrate its potential impact. A more detailed and clear exposition, particularly in the areas highlighted, would significantly strengthen the manuscript’s contribution to the field of urban temperature forecasting using ANN models. I strongly recommend these revisions for the manuscript to be considered ready for publication in Agronomy.
Comments on the Quality of English LanguageModerate editing of English language required
Author Response
Response to Reviewer 1
Thank you very much for taking the time to review this manuscript. Please find the detailed responses below and the corresponding revisions/corrections highlighted/in track changes in the re-submitted files.
The work team has attempted to resolve all of the suggestions, understanding that the changes made improve the research. We believe that your suggestions have been very pertinent and detailed, thank you.
General Comments and Suggestions for Authors
The paper entitled 'Prediction of Daily Ambient Temperature and Its Hourly Estimation Using Artificial Neural Networks in Urban Allotment Gardens and in an Urban Park in Valladolid, Castilla y León, Spain,' submitted for potential publication in Agronomy, delves into a promising field. The use of ANN models for forecasting urban temperatures is indeed a topic of great interest. However, for the manuscript to reach its full potential, I recommend addressing the following points for clarity and completeness.
In its current state, the manuscript, while intriguing, does not fully demonstrate its potential impact. A more detailed and clear exposition, particularly in the areas highlighted, would significantly strengthen the manuscript’s contribution to the field of urban temperature forecasting using ANN models. I strongly recommend these revisions for the manuscript to be considered ready for publication in Agronomy
Specific Comments and Suggestions for Authors
|
Ref. |
Comments and Suggestions for Authors |
Changes and comments made by the authors |
Location of changes in the document |
|
Introduction: |
|||
|
1a |
Reference to 'Significant Attention' (Line 36-39): The claim about the topic receiving significant attention seems unsubstantiated. Could references be provided to support this? |
The concept "significant attention" was modified to "a topic of interest”.
References were added in that sense. |
Available on the lines:
37-38 |
|
2a |
Structure regarding ANNs (Section 1.1): It appears more logical to discuss the broader context of urban temperature variations in Section 1.1 before introducing the specifics of ANNs. |
The previous Section 1.1 has been removed. Its content was transferred to section “1.Introduction”.
In this way, the broader context is described before introducing ANNs specifications. |
Available on the lines:
38-89 |
|
3a |
Temperature Models Beyond ANNs (Section 1.2): The discussion currently focuses narrowly on ANNs. A broader perspective on other methodologies for predicting urban temperatures would enrich the section. |
Now the suggestion is incorporated verbatim into the document:
“In addition, there are also other alternatives used to make urban temperature predictions. For example, IoT, able to predict the citywide surface temperatures with only ambient sensors on board, statistical multivariate methods, which predicts the external temperature value of urban road channels or even remote sensing observation or computer models, used to retrieve the spatial distribution of UHI. Each approach offers unique advantages and considerations for urban temperature prediction.”
|
Available on the lines:
132-138 |
|
4a |
Clear Articulation of Contributions: The manuscript should explicitly state its novel contributions, objectives, and hypotheses, both in the abstract and in the body, to better frame the study's importance and direction. |
Changes were made to the Summary, Discussion, Results and Conclusions with the main objective of writing more clearly the main contributions, importance, and direction of the research. |
|
|
Materials and Methods: |
|||
|
5a |
Justification for Valladolid: The rationale for selecting Valladolid as the study location needs to be clearer. Is there specific meteorological data that underscores its suitability? |
Now the suggestion is incorporated verbatim into the document:
“Valladolid was chosen because research on urban gardens and climate change is being implemented there based on the interest of the University of Valladolid and the Valladolid City Council. This city offers an ideal environment to investigate the impacts of urban allotment gardens and urban parks on urban temperatures with ANNs, because there is a medium-sized city with aspirations in incorporating nature-based solutions to improve their climate adaptation and transform itself into a smart city”
Even a new item has been added in “2.Materials and Methods” (“2.1. The study area”) to better characterize the city and the spaces studied.
This research is part of a more comprehensive investigation on the urban climate. However, it is an investigation in progress and its more specific meteorological data cannot yet be published. |
Available on the lines:
140-151
|
|
6a |
Sensor Location Criteria (Lines 168-169): The decision-making process behind the sensor placements is unclear. An explanation of the criteria used would be helpful. |
Now the suggestion is incorporated verbatim into the document:
“The loggers from the University of Valladolid were Onset HOBO UA-002-64 (5.8 cm × 3.3 cm × 2.3 cm in size, https://www.onsetcomp.com/products/data-loggers/ua-002-64) with an operating range of −20° to 70 °C in air, and an accuracy of ±0.53 °C from 0 to 0–50 °C. The selection criteria for sensor type, quantity, installation height (1.5 meters) and their locations, followed methodologies used, validated and recommended by researchers who are dedicated to investigating the impact of urban gardens on urban temperatures using traditional methodologies [2,3]. In this context, three sensors were deployed in each urban garden (Figure 2) due to the diverse vegetation within the same area and two sensors were placed in Campo Grande (Figure 3), given its more spatially and temporally uniform vegetation (this characteristic potentially results in lower variability of temperature fluctuations at multiple points within the same area).” |
Available on the lines:
205-216 |
|
7a |
Temperature Data Analysis Choice (Line 183): The preference for using average temperatures over maximum temperatures within each hour should be justified |
Now the suggestion is incorporated verbatim into the document:
“The average hourly values of each of the locations were taken into consideration because they provide more representative and stable input data for models training, encouraging better generalization and performance.” |
Available on the lines:
223-226 |
|
8a |
Detailed Description of Urban Allotment Gardens: Given the significant role of vegetation in influencing temperature through factors like albedo and evapotranspiration, a comprehensive description of the urban allotment gardens studied is essential. |
A greater description of urban gardens has been incorporated into the document, based on their main characteristics and vegetation. Information has been added from field sampling on existing crops that helps to understand them.
This information is part of broader research on urban gardens and climate change, therefore, it is considered supporting and complementary material. Available now at: Data Availability Statement |
Available on the lines:
171-198 and in the Data Availability Statement |
|
Results, Discussion, and Conclusion: |
|||
|
9a |
Reorganization of Tables: To enhance readability, consider relocating Tables 1, 2, and 3 to an appendix. The main text should then focus on distilling and discussing the key results. |
Tables 2 and 3 were removed. They are available in Supplementary Materials
The authors consider leaving Table 1 in the main text to visually support the details of the architecture with the best results obtained (6-Y-1) |
Available on the lines:
327-343 |
|
10a |
Vegetation Dynamics and Temperature Influence: The current manuscript does not adequately address how vegetation dynamics influence temperature. A detailed analysis of this aspect, particularly with the limited dataset used, is crucial. |
Now the suggestion is incorporated verbatim into the document:
“The current manuscript does not specifically address how the dynamics of vegetation and urban green areas influence local temperature, but it is part of and contributes to a more comprehensive study where the relationships between urban gardens and climate change are being studied. Therefore, the methodology developed by this study allows the incorporation of innovative techniques to be incorporated into traditional and specific research on the subject.”
This article focuses on highlighting and evaluating the methodology used (ANNs) because it is innovative for this type of study, while the influence of vegetation on urban temperature is part of a more comprehensive investigation that is in progress.
|
Available on the lines:
357-363 |

Reviewer 2 Report (New Reviewer)
Comments and Suggestions for Authors
In the submitted article, the prediction of daily ambient temperature and its hourly estimation in four urban allotment gardens and in an urban park was developed using ANNs. The article is correctly presented, and the subject is interesting. Some minor issues require author's attention:
Page 1, lines 22-23, provide the number of hidden neurons for the developed ANN models.
Page 4, lines 167-176, provide more details about used sensors.
Page 6, specify the activation functions for the inputs and the outputs of the developed ANN models.
Page 13, lines 388-409, highlight the findings of the study without repetition of the aim of the study.
Pages 14-16, use abbreviations for the scientific journals in the reference list.
Author Response
Response to Reviewer 2
Thank you very much for taking the time to review this manuscript. Please find the detailed responses below and the corresponding revisions/corrections highlighted/in track changes in the re-submitted files.
The work team has attempted to resolve all of the suggestions, understanding that the changes made improve the research. We believe that your suggestions have been very pertinent and detailed, thank you.
General Comments and Suggestions for Authors
In the submitted article, the prediction of daily ambient temperature and its hourly estimation in four urban allotment gardens and in an urban park was developed using ANNs. The article is correctly presented, and the subject is interesting...
Specific Comments and Suggestions for Authors
|
Ref. |
Comments and Suggestions for Authors |
Changes and comments made by the authors |
Location of changes in the document |
|
1b |
Page 1, lines 22-23, provide the number of hidden neurons for the developed ANN models |
Now the suggestion is incorporated verbatim into the document:
“Seven ANNs architectures were tested: 7-Y-5 (Y=6,7,…,14), 6-Y-5 (Y=6,7,…,14), 7-Y-1 (Y=2,3,…,8), 6-Y-1 (Y=2,3,…,8), 4-Y-1 (Y=1,2,…,7), 3-Y-1 (Y=1,2,…,7), and 2-Y-1 (Y=2,3,…,8).” |
Available on the lines:
22-24 |
|
2b |
Page 4, lines 167-176, provide more details about used sensors. |
Now the suggestion is incorporated verbatim into the document:
“The loggers from the University of Valladolid were Onset HOBO UA-002-64 (5.8 cm × 3.3 cm × 2.3 cm in size, https://www.onsetcomp.com/products/data-loggers/ua-002-64) with an operating range of −20° to 70 °C in air, and an accuracy of ±0.53 °C from 0 to 0–50 °C. The selection criteria for sensor type, quantity, installation height (1.5 meters) and their locations, followed methodologies used, validated and recommended by researchers…” |
Available on the lines:
205- 210 |
|
3b |
Page 6, specify the activation functions for the inputs and the outputs of the developed ANN models. |
The authors consider that the suggestion was already incorporated in the document:
“The activation function chosen between neurons in the hidden layer was the hyperbolic sigmoid tangent (tansig), while the transfer function for neurons in the output layer was linear (purelin).”
Considerations: The input layer is made up of those inputs, which introduce the values into the network. No processing occurs on these inputs. Although, by convention in the literature, they are represented the same as the hidden and output layers, the inputs are not neurons themselves.
In this architecture we have “by convention” an input layer located on the left that, in reality, does not apply any function to the input values to the network. It is simply the way to indicate that the input values reach the neural network in some way. That is, they are not artificial neurons, although they are represented as such in Figure 4. In the input layer each "neuron" receives a single input value. |
Available on the lines:
270- 272 |
|
4b |
Page 13, lines 388-409, highlight the findings of the study without repetition of the aim of the study. |
Modifications have been made within the discussion and conclusions, in order to make clear the results and the main considerations obtained by the research. |
Available on the lines:
351-439 |
|
4b |
Pages 14-16, use abbreviations for the scientific journals in the reference list |
Most references (especially those scientific journals with long names) were changed to their abbreviations. |
Available on the lines:
460-622 |

Round 2
Reviewer 1 Report (Previous Reviewer 1)
Comments and Suggestions for Authors
Kudos to the authors! Very small things to check, but ready for publication.
Comments on the Quality of English LanguageMinor editing of English language required
This manuscript is a resubmission of an earlier submission. The following is a list of the peer review reports and author responses from that submission.
Round 1
Reviewer 1 Report
Comments and Suggestions for Authors
I have recently reviewed the manuscript titled "Prediction of Daily Ambient Temperature and Its Hourly Estimation Using Artificial Neural Networks in Urban Allotment Gardens and in an Urban Park in Valladolid, Castilla y León, Spain." The paper, submitted for potential publication as a research article in Agronomy, delves into the use of ANN models for urban temperature forecasting. I find the topic intriguing and recognize the potential of the manuscript. However, I have several comments and concerns that need to be addressed for clarity and completeness.
Introduction:
To enhance the flow of the introduction, I recommend structuring it as follows:
Present the current state of the main topic.
Highlight the existing issues or gaps.
Detail how your research aims to address these gaps.
It's crucial to explicitly state your novel contributions, both in the introduction and the abstract. In Section 1.2, please provide necessary references, especially for lines 76-79. I suggest moving lines 89-93 to the end of the introduction to maintain thematic consistency. Section 1.3 seems to blend various methods with ANNs. I recommend reorganizing this section for clarity. Additionally, could you elaborate on other existing methodologies for predicting urban temperatures?
I'm uncertain about the novelty and significance of your study. It would be beneficial if you could elucidate why your research is essential and what sets it apart. Lines 153-157 appear to be conclusions or results and might be better placed in the respective section.
2. Materials and Methods:
Is there additional data available for the ANNs? The data presented seems limited for the scope of a full research paper. Have you calibrated the temperature sensor? It would also be helpful to know the height at which the sensor was placed and the rationale behind it. Regarding lines 226-227, could you explain the choice of six ANN architectures? For lines 231-238, please provide reasoning for selecting this particular configuration. I suggest moving line 247 to the results section.
3. Results:
I recommend being more concise in this section. In my view, six tables seem excessive. Consider summarizing the key findings and relocating all tables to an appendix for reference.
In light of the above comments, I believe the manuscript requires further refinement before it's ready for publication. I encourage the authors to take my feedback into account and resubmit the manuscript for peer review in Agronomy.
Comments on the Quality of English LanguageThe paper requires moderate editing of English language.
Author Response
In response to the reviewer's valuable suggestions, several substantial improvements have been implemented in the manuscript. These enhancements are outlined as follows:
-Introduction Restructuring: The Introduction section has undergone significant restructuring to incorporate the current state of the research topic, elucidate existing challenges, and highlight the noteworthy contributions made by this article.
-Expanded References: Additional references have been incorporated into the bibliography, enriching the paper's scholarly foundation and relevance.
-Enhanced Methodology Description: Efforts have been made to provide concise yet comprehensive explanations of the methodologies employed in this study. These explanations help in facilitating a better understanding and comparison of our work.
-Comprehensive Data Availability: To ensure transparency and reproducibility, all supplementary data, input parameters, and output results generated by various Artificial Neural Networks (ANNs) have been included within the Data Availability Statement annex. This ensures that the employed methodology can be readily followed and replicated.
-Calibration of Temperature Sensors: The temperature sensors, being new, have undergone factory calibration to guarantee their accuracy and reliability in data collection.
-Refined Results and Conclusions: The Results and Conclusions sections have been meticulously revised to enhance their clarity and effectiveness in conveying the key findings and implications of the research.
-Concise Presentation of Tables: Instead of overwhelming the reader with numerous tables, we have streamlined the presentation by including only the three most pertinent ANN tables. For access to the remaining tables, complete with their detailed statistics, readers are directed to the Supplementary Materials, accessible via the provided link.
- Language Refinement: Substantial revisions have been undertaken to enhance the quality of English language usage throughout the article. This ensures that the content is more accessible and understandable to a wider readership.
These revisions and enhancements together contribute to the improved quality and rigor of the manuscript. Your feedback has been invaluable in elevating the overall quality of this scientific article.
Reviewer 2 Report
Comments and Suggestions for Authors
In response to climate threats and the consequences of the urban heat island (UHI) effect, green spaces like urban gardens (allotment gardens in this case) and forested parks are increasingly recognized for their capacity to mitigate these challenges. Obtain of the temperature data from urban locations, however, is challenging. In this study, the authors develops the Artificial Neural Networks (ANNs) to predict daily and hourly data in Valladolid, Spain. The comments are as follows:
Major comment
1. The reasoning of the study seems incomplete. This study picks several places in Valladolid of Spain. However, the choice of study locations in a single city and whether the conclusion can be applied to other cities is not mentioned. I’d suggest the authors discuss the applicability of the current study to other regions. The uniqueness of Valladolid, the choice of time and how studies made here can be applicable to other regions should be mentioned.
2. The creativity of the study. I do not see the creativity in this study, which is usually stated in the introduction. It seems that the authors are trying to state the ANN architecture as creativity, but this does not discriminate from the previous studies using ANN. This should be more stated in the introduction.
3. The language of manuscript should be revised, since it seems there are several language problems in it. The logic flow transition is also weird—one example is the starting sentence of the abstract where cities are mentioned and then sharply turns to urban green space.
4. Are the results compared with algorithms from other studies? This is a way to exhibit the difference of the current study’s model with other previous models, and thus exhibit the creativity of the current study’s model.
Minor comments
Line 1, It seems weird and abrupt logic transition from city to urban green space. The authors may need more edit on transitioning to stating of urban temperature study.
Line 6, “home 295,639 inhabitants”-> “home to 295,639 inhabitants”
Line 24-24, I don’t understand how the “however” is used in this sentence and why other models are mentioned since the authors does not state more about it in the abstract.
Comments on the Quality of English Language
Extensive effort needed for more accurate presentation of the study.
Author Response
In response to your valuable feedback, we have made substantial improvements to the manuscript, addressing your concerns:
- Replicability in Other Cities: We have now included a section discussing the potential for replicating this study in various urban settings. Additionally, we have elaborated on the specific aspects that make our choice of Valladolid as the study location significant.
- Distinctiveness of the Study: We have provided a comprehensive explanation of the unique characteristics that set our study apart from similar methodologies used in urban contexts. This highlights the novelty and relevance of our research.
-Language Refinement: We have thoroughly revised the manuscript to enhance the clarity and precision of the English language used throughout. These changes ensure that the content is more accessible and effectively conveys the research findings.
- Incorporation of Suggestions: We have diligently incorporated all the suggestions you provided, making sure that they are seamlessly integrated into the manuscript to enhance its quality and rigor.
We appreciate your critical review and are confident that these revisions strengthen the scientific merit and readability of our article. Thank you for your continued support and guidance in improving our work.
Round 2
Reviewer 2 Report
Comments and Suggestions for Authors
I feel there are several modifications on the context needed. As follows:
1. Novelty issue. Apparently, the authors still have not been clear on the novelty of the research. In terms of using ANN on urban temperature, I’ve found literature like https://www.mdpi.com/2071-1050/13/15/8143. The authors would have to at least discriminate their studies from studies like this that uses ANN to predict urban temperature. The authors seem to be focusing their studies on urban green space. Yet how this is discriminated from the urban temperature is not discussed at all—this leads to my confusion since using ANN on urban temperature formation is clearly not novel. In terms of the urban green space, the scope seems even smaller than urban temperature. How this is important to the urban temperature and thus be novel would need authors’ greater effort.
2. In terms of comparison with other algorithms. This is to compare with other algorithms to exhibit how this would excel than other used studies. I think there should be other algorithms to compare with that can be utilized in a similar way as ANN. Simply testing the ANN with various configurations does not really tell it. Also, the study period is relatively short (one month training vs one month validation), how the conclusion can be generalized to other regions and other time periods seems to be very limited and questionable.
3. Please provide a manuscript track-change form and state the change in the response. It is really hard to find the changes the authors made in the manuscript.
Comments on the Quality of English Language.